# Gamified M-Health Attention Bias Modification Intervention for Individuals with Opioid Use Disorder: Protocol for a Pilot Randomised Study

**DOI:** 10.3390/ijerph17030752

**Published:** 2020-01-24

**Authors:** Melvyn W. B. Zhang, Sandor Heng, Syidda B. Amron, Zaakira Mahreen, Guo Song, Daniel S. S. Fung, Helen E. Smith

**Affiliations:** 1National Addiction Management Service, Institute of Mental Health, Singapore 539747, Singapore; Sandor_heng@imh.com.sg (S.H.); syidda_amron@imh.com.sg (S.B.A.); zm_shahul_hameed@imh.com.sg (Z.M.); song_guo@imh.com.sg (G.S.); 2Institute of Mental Health, Singapore 539747, Singapore; Daniel_fung@imh.com.sg; 3Family Medicine and Primary Care, Lee Kong Chian School of Medicine, Nanyang Technological University Singapore, Singapore 308232, Singapore; h.e.smith@ntu.edu.sg

**Keywords:** attention bias, cognitive bias, addiction, psychiatry

## Abstract

**Introduction**: Globally, there is an epidemic of opioid use disorders. Locally, in Singapore, there is an increase in the number of individuals abusing opioids. The advances in experimental psychology have highlighted the need to modify unconscious, automatic biases. These automatic, unconscious biases result in individuals having preferential attention to substance-related cues in their natural environment, thus leading to a slip or relapse back into their underlying addictive disorders. Prior studies have demonstrated not only the presence of robust attentional biases amongst individuals with opioid use disorder, even when maintained on methadone; and the effectiveness of bias modification amongst these individuals. There remains limited evaluation of attention bias modification amongst a treatment-seeking cohort of Asian individuals. The objective of this pilot is to ensure that the methods of the planned definitive randomized trial could be conducted. **Methods and Analysis**: A non-blinded pilot randomized trial will be conducted. A total of 30 participants will be randomized to receive either the conventional application or the newly designed co-designed application. In order to identify these 30 participants, 60 participants will be recruited and screened to determine if they have baseline biases. Participants will be recruited from the inpatient unit at the National Addictions Management Service (NAMS) Singapore. All participants who are enrolled into the trial will complete a baseline assessment task, and a bias modification assessment and modification task daily. They will have to complete a baseline demographic and clinical information questionnaire, as well as a cravings rating scale before and after the intervention daily. Perspectives—that of self-reported experiences—will be sought from the participants following their completion of three intervention tasks. Descriptive statistical analyses will be performed, and chi-square and ANOVA analyses will be performed. Qualitative analyses will be undertaken for the perspectives shared. **Ethics and Dissemination**: Ethical approval has been obtained from the National Healthcare Group’s Domain Specific Research Board (DSRB) (approval number that of 2019/00934). The findings arising from this study will be disseminated by means of conferences and publications.

## 1. Introduction

Substance use disorders are becoming increasingly prevalent. In 2015, the United Nations Office of Drugs and Crime (UNDOC) estimated that at least a quarter of a billion individuals have experimented with substances, and 0.06% of the global adult population (or 29.5 million individuals) were diagnosed with a substance use disorder [1]. According to the latest report released by the UNDOC, there has been an increase since then, with an estimated 35 million individuals diagnosed with substance use disorder [1]. The UNDOC’s latest report also highlights there being a treatment gap, given that only one in every seven individuals have received treatment. In 2015, the most common substances that were abused were that of cannabis, opioids and amphetamines, with annual prevalences of these of 3.8%, 0.7% and 0.77% respectively, each year [1]. In 2019, there has been a large increase in the number of opioid-using individuals; in fact, a 56% increment from previous estimates [1]. It is well-known that substance use disorders are associated with significant morbidities and mortality. With regards to mortality, two thirds of individuals diagnosed with substance use disorders who died in 2017 were opioid users [1]. With regards to morbidities, an estimated 1.4 million substance-using individuals are diagnosed with HIV (human immunodeficiency virus), and another 5.6 million individuals with hepatitis C [1].

Globally, there is an epidemic of opioid use, according to the World Health Organization. In 2016, an estimated 27 million individuals were diagnosed with opioid use disorders. Most of these individuals have abused illicit opioids, but an increasing number are using prescription opioids, such as tramadol and oxycodone [2]. Several factors are responsible for the current epidemic of abuse of prescription opioids, and they include that of the increased accessibility to prescription opioid analgesics, the enhanced purity of heroin and the introduction of more potent illicit fentanyl compounds [3,4]. Several public health approaches have been considered, such as that of prescribing only the right quantities, heightened measures to prevent diversion and changes in drug formulation in order to prevent abuse [3,4]. Whilst these approaches are essential, there needs also to be efficacious treatment for opioid use disorder. Opioid substitution medications like methadone (opioid receptor agonist) and that of buprenorphine (partial opioid receptor agonist) are routinely used. Psychosocial treatment is also routinely coupled with pharmacological treatment, to help in harm minimization and eventual abstinence. 

In Singapore, there remains no recent large-scale epidemiological study that has been conducted for substance use disorders. However, in 2004, the World Health Organisation estimated the prevalence of substance use disorders in females to be 0.07% and in males to be 1.40% [5]. Statistics from the Central Narcotics Bureau Singapore (2018) have revealed that heroin was among the top three most commonly abused drugs apart from that of methamphetamine and novel psychoactive substances (NPS) [6]. Traditionally pharmacological medications, that of opioid substitutes, have been recommended in conjunction with psychosocial interventions. However, in Singapore, opioid substitutes are not available, due to Singapore’s zero tolerance for drug use policy. Instead, individuals with opioid use disorders undergo detoxification using symptomatic medications, followed by psychological interventions [7]. 

In the Singapore context, psychosocial interventions are of particular importance, given the limited pharmacological options that are available. Psychological therapies help individuals to achieve and maintain abstinence. Traditionally, therapies like cognitive behavioural therapy for relapse prevention, mindfulness-based relapse prevention and cue-exposure therapies are used. Whilst these therapies, such as cognitive behavioural therapy, have been proven to be effective for disorders including alcohol disorders, in some studies, a proportion of individuals still relapse following intervention [8]. It is likely that these therapies have addressed only the cognitive control processes leading someone to lapse or relapse into an addictive disorder, but not the underlying unconscious processes [9]. The advances in experimental psychology have led to the greater understanding of these unconscious, automatic biases—that of attention and approach biases [9,10]. The dual-process theoretical approach states that attention biases (preferential allocation of attention to substance-related cues) and approach biases (automatic action tendencies to reach out for substance-relate cues) develop as the repetitive usage of a substance leads to increased automatic processing and automatic tendencies to acquire the substance, with the inhibition of the normal cognitive control processes [10,11]. Maclean et al. (2018) [12], in their prior review, highlighted there being robust attentional biases present amongst individuals with opioid use disorders. This was based on the evidence synthesized from 21 studies. Maclean et al. (2018) [12] also reported there being a moderate to large effect size for the presence of attentional biases amongst individuals with opioid use disorders, as compared to healthy controls. With regard to attention bias modification, studies like that of Hietmann, J. et al. (2018) [13] in their systematic review, which included nine studies on alcohol use disorder, six studies on nicotine use and three on opiate use disorders, reported that whilst there have been negative findings, multi-session attention bias modification has had positive effects on attentional change, particularly for alcohol use disorders. Ziaee et al. (2016) [14] reported bias modification to be effective in reducing attention bias for drug-related stimuli and temptations to use, amongst individuals maintained on methadone maintenance therapy. However, the fact that attentional biases are still persistent despite being on methadone maintenance therapy also highlights the need for specific intervention to address these unconscious attentional processes [15]. Most of these studies are limited to the evaluation of attentional biases in western individuals, and to date, from our knowledge, there remains only a single study (Zhang et al., 2019) [15] that has explored attentional bias modification in an Asian cohort of substance-using individuals. In Zhang et al. (2019)’s [15] prior study, they reported the feasibility and acceptability of a mobile attention bias intervention and reported there being individuals without baseline biases. In their sample of 30 participants, 17 participants had a diagnosis of opioid dependence, of which eight had an absence of baseline biases [16]. Zhang et al. (2019)’s [15] prior study did demonstrate that there was a change amongst participants with baseline biases, and this warrants further exploration. 

Zhang et al. (2019) [17] subsequently undertook a co-design study and involved both healthcare professionals and patients in the joint conceptualization of a new intervention to address the shortcomings of the conventional application, particularly that of motivation to continue training and the inherent repetitiveness of the task. Both groups of participants suggested for there to be a lengthier stimulus presentation interval at the start of the intervention, before gradual decreasing to a short presentation interval. Gamification strategies, mainly that of feedback, were recommended. To date, Zhang et al. (2019) [17] remains the first study to have involved participants in the design of an attention bias modification intervention, and there needs to be further evaluation of the co-design application against that of the original application to determine if it is more effective. 

The objective of the current pilot randomized trial is to determine if the intended processes of the definitive study (i.e., the recruitment of participants with opioid use disorder, the process of randomization into two separate intervention arms, and the administration of the intervention) would work. The current pilot differs from the actual definitive randomized trial, in that the intent is essentially in testing out the process for the definitive trial. This pilot is of importance, as it will help identify and eliminate problems that might affect the eventual success of the definitive trial. Prior research has highlighted how as much as 85% of research investment has been wasted due to the poor design and conduct of trials [18]. Some of the previously identified issues that render a trial to be poor include a lack of systematic literature review and inadequate concealment of treatment allocation [18].

## 2. Methods

### 2.1. Study Design and Setting

The study design will be that of a non-blinded randomised controlled trial. Participants will be allocated to receiving either the conventional attention bias modification task, or that of the co-designed task. Simple randomisation will be used. The study will be conducted on the inpatient unit at the National Addictions Management Service, Institute of Mental Health, Singapore. This is a non-blinded randomised trial, as participants who are allocated to both the groups will knowingly be aware of their treatment allocation. It is also not practical for study team members to be blinded, as they are to be responsible for the group allocation, intervention and data collation. 

### 2.2. Inclusion and Exclusion Criteria

Participants are considered eligible for the study if they fulfil the following inclusion criteria, (a) aged between 21 and 65 years old; (b) primary psychiatric diagnosis of opioid use disorder; (c) for patients diagnosed with polysubstance use disorder, their primary substance of abuse needs to be that of opioids; (d) able to read and write in English; and (e) completed their detoxification program. For participation in the trial, participants are required to have baseline attentional biases (i.e., have a positive attentional bias).

### 2.3. Sampling and Recruitment

Convenience sampling will be used in the recruitment of participants over a period of 12 months, at the inpatient unit of the National Addictions Management Service (NAMS), Institute of Mental Health Singapore. The inpatient unit at NAMS offers a 2-week program, in which the first week of the program is focused on the provision of medication-assisted detoxification and the second week of the program is focused on rehabilitation. The unit can have a total of 22 inpatients at any one time. It is also crucial to note that the program is voluntary in nature, which implies that participants are free to leave the ward if they wish to, at any juncture of their inpatient treatment. All participants who have demonstrated interest in the study will be asked to undertake a baseline attention bias assessment task. Only participants who have baseline biases will be recruited into the trial and will be randomly assigned to any of the two groups by means of simple randomisation. Figure 1 provides an overview of the participant flowchart following screening and recruitment.

### 2.4. Sample Size

For the purposes of this pilot, we plan to screen 60 participants for baseline biases. Based on our prior feasibility and acceptability study [15], approximately 50% of these participants will have baseline biases. We anticipate that 50% of the participants (or 30 participants) may not have baseline attentional biases and hence would not be eligible for the eventual pilot. Therefore, the sample size for the eventual pilot is that of 30 participants, with 15 participants being randomised to receiving the co-design application and the remaining 15 being randomised to receiving the conventional applications. The sample size of 30 participants in total, with 15 in each group, was selected, taking into consideration previous pilot randomized trials evaluating web-based cognitive bias interventions for individuals with obsessive-compulsive disorder [19,20]. In those previous studies, the sample size per group ranged from between 6 and 37 participants. Moreover, based on Bell LM et al. (2018)’s [21] guidance for using pilot studies in informing the design of confirmatory studies, the proposed sample size for the pilot study if the effect size is small (i.e., Cohen D is between 0.1 and 0.3), is an estimated 25 participants per arm of the intervention. We have not performed a power computation for the purposes of the pilot, as the pilot study is supposed to inform the definitive sample size for the confirmatory trial [22,23]. 

### 2.5. Randomisation

Simple randomisation, by means of a computer program, will be used, and each participants’ treatment will be determined independently, without any constraints. In this study, the method proposed by SD Simon (1999) [24] will be used. Making use of Microsoft Excel, a spreadsheet will be created, with a column indicating the participant number of 60 participants. Another column will be created in which the intended treatment will be arranged systematically. Using a column of random numbers, by using the RAND () functionality, the column comprising of the random numbers will be sorted, which will in turn result in the treatment conditions being allocated to participants in a random and unpredictable order [25]. 

## 3. Experimental Procedure and Data Collation

**a.** 
**Screening for Eligibility**


Informed consent will be acquired from participants on the last day of their detoxification stay (Day 7), for the purposes of screening, in the presence of an impartial witness. For the screening for the presence of baseline attentional biases, participants are required to undertake an attention bias task on a tablet device. In the assessment task, participants will be shown a fixation cross for 500 milliseconds, followed by a set of images (one related to their substance of use and another neutral image that is similar in complexity). While 50% of these images will be presented for a short stimulus duration of 200 milliseconds, the other 50% of the set of images will be presented for a long stimulus duration of 2000 milliseconds. Following the disappearance of the images, participants are presented with a probe (in the form of an asterisk) for 200 milliseconds. Participants are required to indicate their response, by means of tapping the buttons on the screen, to indicate if the asterisk is presented on the left or right side. If they do not respond, they will be presented with the next trial. In the screening phase, 50% of the trials (100 trials) will involve the presentation of the probe with the substance image, and 50% of the remaining trials will involve the presentation of the probe with the neutral image. The presence of attentional bias will be computed by means of subtracting the reaction time that participants take to respond to probes that replace the neutral image from the reaction time to the probe that replaced the substance image. 

**b.** 
**Experimental Procedure**


Participants who are screened positive (i.e., having a positive baseline biases) will be eligible to be recruited into the pilot randomized trial. Informed consent from these participants will have already been obtained in the screening stage. Participants who are ineligible (those without baseline biases) will be dismissed from the study, given due compensation and allowed to go on with their rehabilitation program. Members of the study team will orientate the participants on the use of the mobile application. Participants will be provided with a tablet device to use the mobile application daily. Participants will be also provided with a unique username and password to allow them to have access to the application daily. Participants will be required to complete the following baseline questionnaires, the baseline demographic and clinical information questionnaire, on Day 8 of their stay in the ward, or the first day of their intervention. The following baseline demographic and clinical information will be collected from the participants—that of nationality, gender, marital status, race, religion, highest level of education, housing conditions, current substance use, method of consumption of substance, quantity of substance consumed each time, frequency of use, previous treatment history, chronic diseases (psychiatric and physical disorders) and current psychiatric medications. Participants will also be required to complete a visual analogue scale for the assessment of cravings before and after undertaking the intervention. The experimental procedure will not differ for participants who are randomized to receiving the co-designed application versus that of the conventional application. Participants will be asked to spend 10 min to complete 200 repeats of the intervention task. Upon completion of the first intervention task, participants can rest for 10 min, before spending another 10 min to complete 200 repeats of the assessment task. Participants will continue to retake the intervention daily, each day they are an inpatient, except for weekends and public holidays. On the subsequent days, participants are to take the intervention task, rest for 10 min, before completing an assessment task. Participants will be asked to retake the craving questionnaire prior to and after the tasks on each of the days that they are involved in the study. Participants will be also asked to complete a questionnaire to acquire their feedback pertaining to the intervention (after they have completed three interventions). If participants request premature discharge from the ward, they are not required to return to the ward to complete the intervention. 

Participants will also be asked to complete a perspective/self-reported questionnaire on Day 3 of the intervention (i.e., when they have completed a total of three intervention tasks). This questionnaire has previously been used in our feasibility and acceptability study [15]. Participants will be asked the following questions:Prior to using the application, how confident were you in managing your addiction problem? (5-point Likert Scale)How easy was it to use the application? (5-point Likert Scale, ranging from Not at all to extremely, and written comments)How engaging was the application? (5-point Likert Scale, ranging from Not at all to extremely, and written comments)Do you feel motivated to continue to use the app? (5-point Likert Scale, ranging from Not at all to extremely, and written comments)Do the images in the app remind them of their substance use? (5-point Likert Scale)After using the app, how confident are you in managing your addiction problem? (5-point Likert Scale)

Figure 2 provides an overview of the study schedule/data collection time points and tasks participants are required to undertake. 

**c.** 
**Details of Intervention Tools**


For the conventional mobile attention bias modification application, there are two tasks that participants are to undertake daily—that of the assessment task and that of the intervention task. In the assessment task, participants are presented with a fixation cross at the centre of the screen for 500 milliseconds. The fixation cross will then disappear, and a set of images will be presented. 50% of the set of images (i.e., 100 sets of images) will be presented for 200 milliseconds, and 50% of the set of the images (i.e., 100 sets of images) will be presented for 2000 milliseconds. The set of images will comprise a substance image and a neutral image, that is similar in colour, size and complexity. The images that will be used will be obtained from the Central Narcotics Bureau’s database. Following the disappearance of the images, a probe (asterisk) will appear, either replacing the substance image or the neutral image, for 200 milliseconds, before disappearing. Participants are supposed to indicate the position of the probe by selecting the buttons on the screen (i.e., left button if the probe is on the left side, right button if the probe is on the right side). In the assessment task, 50% of the time, the probe will replace the substance image, and in the remaining 50% of the time, it will replace the neutral image. Participants are to complete 200 repeats in the assessment task. There are 10 sets of images, which will be repeated 20 times over the 200 repeats. The time interval between each repeat is 250 milliseconds. In the intervention task, the procedure is like that of the assessment task, with the exception that the probe will always replace the neutral image (100% of the time). This allows for the retraining of biases away from the substance image, and towards that of the neutral image. Participants are to also complete 200 repeats during the intervention task. 

For the co-designed intervention, there are also two tasks that participants are to undertake daily, that of the assessment and that of the intervention task. The application contains an instructional video to help participants better understand what they would need to do in these tasks. Participants are to complete 5 sample trials before they undertake the assessment or intervention task. In the assessment task, participants are presented with a fixation cross at the centre of the screen for 500 milliseconds. The fixation cross will then disappear, and a set of images will be presented. The duration for which the images stay on the screen will vary across the days of the intervention. Table 1 provides an overview of the timings of the images by days of the intervention. 

The set of images will comprise of a substance image and a neutral image which is similar in colour, size and complexity. The images that are used will be obtained from the Central Narcotics Bureau’s database. Following the disappearance of the images, a probe (asterisk) will appear, either replacing the substance image or the neutral image for 200 milliseconds, before disappearing. Participants are supposed to indicate the position of the probe by selecting the buttons on the screen (i.e., left button if the probe is on the left side, right button if the probe is on the right side). If participants responded correctly, they will be presented with a smiley emoji. If they have not responded correctly, they will be presented with a frowning emoji. The scores that they have accumulated across the trials will be presented on the top right-hand corner of the application. Participants are to complete 200 repeats in the assessment task. There are 10 sets of images, which will be repeated 20 times over the 200 repeats. Participants will be presented with their final scores upon the completion of the task. The time interval between each repeat is 250 milliseconds.

In the intervention task, the procedure is like that of the assessment task, but with the exception that the probe will now always replace the neutral image, to allow for retraining of biases away from the substance stimulus. 

## 4. Outcomes

The main outcome that is being evaluated in this pilot study is whether the intended number of participants could be recruited, within the duration of one year. This pilot study also seeks to determine if the intended processes of randomization and the administration of the intended interventions would be successful. In this current pilot study, we acknowledge that selection biases are present, in that we are selecting a cohort of participants that voluntarily seek admission to the inpatient unit. Moreover, the administration of the screening test would also imply that we are selecting specifically for individuals with positive baseline biases only. We are of the opinion that such biases could not be reduced and removed, and we will acknowledge them in our final published study. We will also acknowledge that such biases might affect the overall generalizability of our results. 

### 4.1. Statistical Analyses

For data analysis, an intention to treat analytical approach will be used, to include all participants, including those who withdrew and those who are prematurely discharged from the inpatient unit. Microsoft Excel will be used for the computation of the attention bias scores for each participant for each of the assessment task, in accordance to this formula:(∑T_1_/n_1_) − (∑T_0_/n_0_),
where 

T_1_ refers to the time for probes that replaced the neutral stimulusn_1_ refers to number of trials for probes that replaced the neutral stimulusT_0_ refers to the time for probes that replaced the substance stimulusN_0_ refers to number of trials for probes that replaced the substance stimulus

SPSS Version 24.0 (SPSS Inc., Chicago, IL, USA) will be used for the analysis of the quantitative data. Demographic and clinical information data of the subjects in both groups will be summarized using descriptive statistics, including that of means and standard deviations. Chi-square tests will be used to determine if there are differences in the two groups for each of the demographic or clinical information variables. We plan to run ANOVA (analysis of variance) analysis to examine the differences in the outcome measure (attention bias scores/reaction time) across the time points in the actual definitive trial. P-values less than 5% will be considered statistically significant. 

For the qualitative data obtained by means of the perception questionnaire, the data will be coded using NVivo 12, and a thematic analysis approach will be used in the analysis. 

### 4.2. Data Management and Monitoring

In order to protect the confidentiality of the participants and for the data to be de-identified, all the participants will be allocated a unique subject number upon recruitment. All the hard copy questionnaires will contain the unique subject number and no other participant-related identifiers. All the informed consent forms and the completed hard copy questionnaires will be stored in secured, locked cabinets in restricted areas. The electronic data (which comprise mainly reaction times of the participants) will be automatically synchronized to a cloud database in Singapore (Digital Ocean, Singapore). A copy of the electronic data will be downloaded daily onto a local secured computer and stored onto a research network drive. The password of the local computer will be changed frequently. The study team members will code the data and an independent co-investigator will routinely check the data coding for reliability and quality. All the hard copy forms and electronic data will be kept for at least six years following the completion of the study. 

**Adverse Events:** Any adverse events that occur during the conduct of the pilot study will be reported to the Domain Specific Research Board in accordance with the local institutional policy.**Ethics:** The protocol for this study has been approved by the National Healthcare Group Domain Specific Research Board (DSRB Reference Number: 2019/00934). **Expected Results and Conclusions:** This pilot study will provide valuable information about whether the intended processes for the definitive trial would work. This study is of importance, as it will help guide the actual definitive randomised trial. Recruitment and data-analysis is expected to be completed in a duration of 12 months. The results obtained from this study will be disseminated by means of local and international conferences, and by means of peer-reviewed publications.

## Figures and Tables

**Figure 1 ijerph-17-00752-f001:**
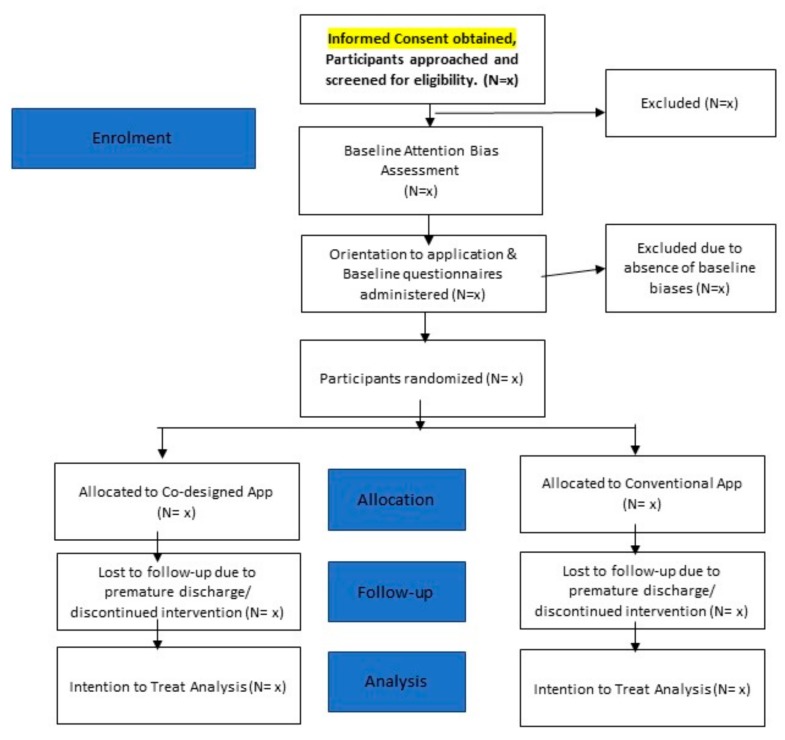
Overview of Participant Flowchart.

**Figure 2 ijerph-17-00752-f002:**
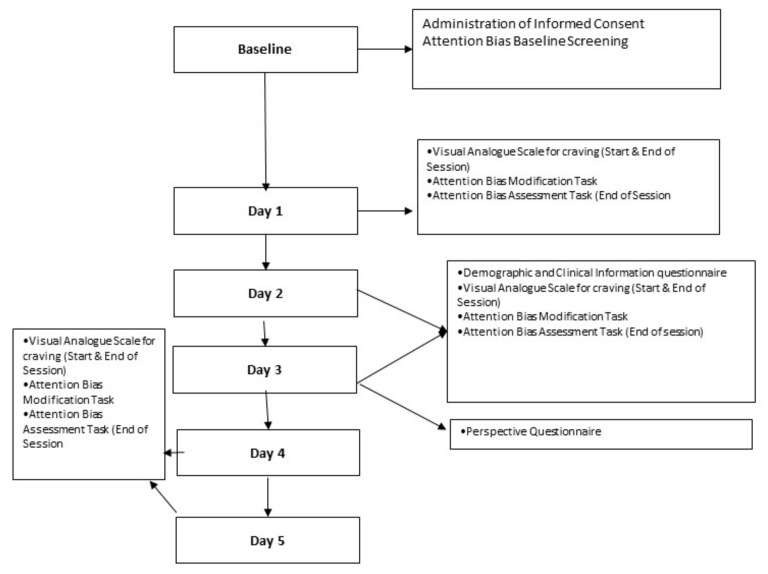
Overview of the data collection time points and tasks to be undertaken.

**Table 1 ijerph-17-00752-t001:** Overview of timings of the images by the days of intervention.

Day of Intervention	Total Number of Trials	Percentage of Image Sets Presenting for 200 Milliseconds	Percentage of Image Sets Presenting for 2000 Milliseconds
Day 1	200	10%	90%
Day 2	200	20%	80%
Day 3	200	30%	70%
Day 4	200	40%	60%
Day 5	200	50%	50%

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
