# Peer review of "Gamified M-Health Attention Bias Modification Intervention for Individuals with Opioid Use Disorder: Protocol for a Pilot Randomised Study"

_ijerph, 2020, doi:10.3390/ijerph17030752_

Round 1

Reviewer 1 Report

Please, see attached file!

Author Response

Replies to Peer Reviewer 1.

We thank you for peer reviewing our manuscript and for your kind comments and recommendations. Please find as appended our comments and our revisions to our manuscript.

This is an interesting and well-designed pilot study on gamified M-health attention bias modification in opioid use. Treatment of attention bias in addiction is an interesting option, which needs exploration. The manuscript is basically well written.

We thank you for recognizing the importance of this work and for recognizing that targeting attentional biases for individuals with addictive disorders is a viable treatment option.

However, one of the aims in the abstract, namely the outcome of co-design is never followed up in the paper.

We like to clarify that in this pilot, the objective is only in determining if the intended processes for the RCT would work. We are not evaluating if the codesigned application will lead to there being a greater reduction in the overall attentional bias, as compared to the conventional application, as the current study is not adequately powered to do so.

The gambling intervention could be somewhat more clearly described, as the other parts of method already is. An international reference outside Asia should be mentioned, i e Heitman et al Systematic review 2018, BMC, not only Zhang.

Thank you for your suggestion. I have included this reference and modified our manuscript. Please find the revisions as enclosed: “With regards to attention bias modification, studies like that of Hietmann J et al. (2018) in their systematic review, which included 9 studies on alcohol use disorder, 6 studies on nicotine use and 3 on opiate use disorders, reported that whilst there have been negative findings, multi-session attention bias modification have had positive effects on attentional change, particularly for alcohol use disorders. Ziaee et al. (2016) (12) have reported bias modification to be effective in reducing attention bias for drug-related stimuli and temptations to use, amongst individual maintained on methadone maintenance therapy. However, the fact that attentional biases are still persistent despite being on methadone maintenance therapy also highlights there being a need for specific intervention to address these unconscious attentional processes (13).”

More specifically: Line 51 co- design… this aim is not mentioned in the last paragraph (as mentioned above)

Thank you for highlighting this. We have revised and specified the aims more clearly. We have made amendments to the aims pertaining to the outcome of the co-design, i.e. we have removed the aim of assessing for difference given that the current pilot is not designed to be adequately powered to detect a statistically significant difference in the mean attention bias scores.

Line 71 and 73, 29.5 billions to 35, I suggest the authors to insert ’increased’ as two years are compared, to make it more readible.

We have included your recommendations. The amends are as follows:
“In the latest report released by the UNDOC, there have been an increase, with an estimated 35 million individuals diagnosed with substance use disorder”

79-80 two thirds of what??? This is not clear.

We have amended this sentence as follows: “With regards to mortality, two thirds of individuals diagnosed with substance use disorders who died in 2017 were opioid users (1).”

94 the new data for Singapore could be compared with the previous ones, previous page, worldwide.

Thank you for this suggestion. However,we do not have the percentages of individuals in Singapore diagnosed with each of the different types of substance use disorders, and hence we are unable to provide a direct comparison.

Further description is easy to follow. 132 other ref apart Zhang and Asia, from the rest of the world (see above!)

We have included a recent systematic review, as per your suggestion, that was conducted by Hietmann J et al. (2018) about the effectiveness of attention bias modification for substance use disorders.

174 aims are missing?? Please, add!

Thank you for highlighting this. We have made the following amends:

“The objective of the current pilot randomized trial is in determining if the intended processes of the definitive study (i.e. the recruitment of participants with opioid use disorder, the process of randomization into two separate intervention arms, and the administration of the intervention) would work. The current pilot differs from the actual definitive randomized trial, in that the intend is essentially in testing out the process for the definitive trial. This pilot is of importance, as it will help identify and eliminate problems that might affect the eventual success of the definitive trial. Prior research has highlighted how as much as 85% of research investment has been wasted due to the poor design and conduct of trials (19). Some of the previously identified issues that render a trial to be poor include a lack of systematic literature review and inadequate concealment of treatment allocation (19).”

208 …pilot is… (add ’therefore’ to make it more readible) Did 60 fall out?

We have added the terminology “therefore” as per your suggestion.

The authors expected that 30 out of 60 would have an attentional bias. Was it so or 59, 61, whatever?? I wonder.

We have clarified this. We stated that we anticipate that 50% of the participants (or 30 participants) may not have baseline attentional biases and hence would not be eligible for the eventual pilot. Thus, out of the 60 participants, 30 participants might not go onto the eventual trial. We have clarified that we seek to recruit 30 participants, with 15 participants being randomized to receive the co-design application and 15 being randomized to the conventional application.

378 Expected… the outcome of the pilot does not seem to be ready yet. I would prefer a submission of a complete pilot study, both aims being answered.

We would eventually submit the results of our pilot study, when it has been executed. For now, we seek the publication of our planned protocol for our pilot study.

Reviewer 2 Report

Thanks for the opportunity to review the present paper! 

While the study subject is potentially interesting and clinically relevant, I have a few concerns which - in my opinion - have to be addressed. 

Abstract

Several parts of the abstract needs to be clarified and improved. The wording 'automatic biases' does not tell the authors what this is really about - this phrase could be understood as any biases such as biases in research etc. This should be better written. 

Here, the actual study concerned should be better outlined. is this the abstract describing the pilot study, the main study or both? 

'if the co-designed application is more effective' isn't enough as description of the aim of testing an intervention. It needs to be clearer in the abstract what this study actually wants to test. Consider omitting the more general information provided in the beginning of the abstract instead. 

Here, the authors state that 30 (out of 60) patients will be randomized. As this is an RCT, readers might think all 60 are included in two arms (30+30), which I understand is not the case. Please revise this section and improve clarity here. Readers who read only the abstract should still understand the basic procedures of the study.

Convenience sampling - what does that refer to here? It is unclear in the abstract how it is related to the 60 patients included, the 30 (approx.) randomized, etc. Please explain.

'Perspectives will be sought...' is not a very distinct wording. Please revise. Self-reported experience or something similar? 

Qualitative thematic analyses. Which are these? They do not seem to be very well explained in the main text either. Is an actual qualitative sub-study included in this protocol? 

Background 

The overall research interest is obvious but needs to be better outlined by the introduction section. Now, it describes a great deal of background relevant only to the clinical setting of Singapore specifically, although the rationale behind the study is more closely related to the mechanisms of addiction to opioids and potentially applicable to any setting. Likewise, the background section needs to better separate what is relevant to licit and illicit opioid use, to describe the opioid problem more generally and not US-specifically. This section would merit from being shorter and more focused. 

Retroviral disease seems to be an outdated way of referring to HIV.

The aims of the study are stated at the end of the introduction but can seem misleading to the reader. The authors should put some effort into making the aims completely unambiguous to the readers. Most importantly, the different between the pilot and the main study needs to be very clear. 

The very last paragraph of the introduction - which aims to justify why a pilot study is conducted - could be better integrated into the text instead of appearing as a separate paragraph at the end. 

Methods

The lower age limit (21 years) should be explained. 

A power calculation has to be included, and needs to be of relevance both to the pilot study and to the main study. 

In the randomization section, the authors refer to the work of the author 'Simon S'. Is that name correct?

It is difficult to understand whether participants will be informed and included at the first assessment or at the end of the detox period? Will the baseline measure take place prior to informed consent? The exact time schedule should be clarified. 

The authors may need to justify why religion is assessed as one of the variables collected from patient, or consider removing this from the study. To me, it is of unclear relevance, unless the authors argue otherwise. 

The authors should consider naming the 'perspective questionnaire' in a better way. Is this a questionnaire measuring self-reported or subjective patient experience, patient satisfaction or similar? Is this questionnaire based on any previous study, any previous data or an evidence-based measure? Please explain this. 

The authors refer to a qualitative sub-study, but as I understand it, it is not included in the present study protocol? Please clarify this. 

Author Response

Replies to Peer Reviewer 2

We thank you Peer Reviewer 2 for reviewing our manuscript. You have provided us with valuable inputs and recommendations. We have amended our manuscript substantially, taking into consideration your recommendations.

Thanks for the opportunity to review the present paper! 

While the study subject is potentially interesting and clinically relevant, I have a few concerns which - in my opinion - have to be addressed. 

Abstract

Several parts of the abstract needs to be clarified and improved. The wording 'automatic biases' does not tell the authors what this is really about - this phrase could be understood as any biases such as biases in research etc. This should be better written. 

Thank you for highlighting this. We have appended more information, as follows “These automatic, unconscious results in individuals having preferential attention to substance-related cues in their natural environment, thus leading to a slip or relapse back into their underlying addictive disorders.”

Here, the actual study concerned should be better outlined. is this the abstract describing the pilot study, the main study or both? 

We seek to clarify that the abstract is describing the intended processes of the pilot study. We have stated in the abstract that “a non-blinded pilot randomized trial” will be conducted.

'if the co-designed application is more effective' isn't enough as description of the aim of testing an intervention. It needs to be clearer in the abstract what this study actually wants to test. Consider omitting the more general information provided in the beginning of the abstract instead. 

We thank you for your suggestion. We have clarified the intents of our pilot, as follows, “The objective of the current pilot randomized trial is in determining if the intended processes of the definitive study (i.e. the recruitment of participants with opioid use disorder, the process of randomization into two separate intervention arms, and the administration of the intervention) would work. The current pilot differs from the actual definitive randomized trial, in that the intend is essentially in testing out the process for the definitive trial. This pilot is of importance, as it will help identify and eliminate problems that might affect the eventual success of the definitive trial. Prior research has highlighted how as much as 85% of research investment has been wasted due to the poor design and conduct of trials (19). Some of the previously identified issues that render a trial to be poor include a lack of systematic literature review and inadequate concealment of treatment allocation (19).”

It is not the objective of this pilot in determining if there was a statistically significance difference in the mean reduction of the attention bias scores following the administration of the co-designed application, as compared to the conventional applications.

Here, the authors state that 30 (out of 60) patients will be randomized. As this is an RCT, readers might think all 60 are included in two arms (30+30), which I understand is not the case. Please revise this section and improve clarity here. Readers who read only the abstract should still understand the basic procedures of the study.

Thank you for this suggestion. We have clarified this as follows: “A total of 30 participants will be randomized to receive either the conventional application or the newly designed co-designed application. In order to identify these 30 participants, 60 participants will be recruited ad screened, to determine if they have baseline biases.”

Convenience sampling - what does that refer to here? It is unclear in the abstract how it is related to the 60 patients included, the 30 (approx.) randomized, etc. Please explain.

We have removed this from the abstract so as to minimize any confusions.

'Perspectives will be sought...' is not a very distinct wording. Please revise. Self-reported experience or something similar? 

Thanks for your suggestion. We have revised this to: ‘Self-reported experience”

Qualitative thematic analyses. Which are these? They do not seem to be very well explained in the main text either. Is an actual qualitative sub-study included in this protocol? 

We seek to clarify that we intend to conduct a qualitative analysis of the verbatim comments offered by the participants in the perspective questionnaire. We have revised the abstract to state that “qualitative analyses will be undertaken for the perspectives shared.

Background 

The overall research interest is obvious but needs to be better outlined by the introduction section. Now, it describes a great deal of background relevant only to the clinical setting of Singapore specifically, although the rationale behind the study is more closely related to the mechanisms of addiction to opioids and potentially applicable to any setting. Likewise, the background section needs to better separate what is relevant to licit and illicit opioid use, to describe the opioid problem more generally and not US-specifically. This section would merit from being shorter and more focused. 

We thank you for your recommendation. We have intentionally described the background relevant only to the clinical setting of Singapore, as Singapore has a unique opioid treatment policy. In Singapore, methadone maintenance therapy is not available, and participants routinely undergo detoxification using symptomatic medications. Given this, there is hence a need for efficacious psychological interventions, such as that of attention bias modification. It is true that attention bias modification would work for addiction for opioids in any clinical setting. We have not discounted this in our introduction. In our introduction, we have stated explicitly that the opioid epidemic has been contributed by both the abuse of the illicit and prescription opioids. We have tried to describe the opioid problem more generally. We have included the following amends, “ Globally, there is an epidemic of opioid use, according to the World Health Organization. In 2016, an estimated 27 million individuals were diagnosed with opioid use disorders. Most of these individuals have abused illicit opioids, but an increasing number are using prescription opioids, such as tramadol and oxycodone (3). Several factors are responsible for the current epidemic of abuse of prescription opioids, and they include that of the increased accessibility to prescription opioid analgesics, the enhanced purity of heroin and the introduction of more potent illicit fentanyl compounds (2,3).” We have also shortened the introduction by removing unnecessary information and repetitive information.

Retroviral disease seems to be an outdated way of referring to HIV.

We have removed the terminology and have used the term “Human Immunodeficiency Virus” instead.

The aims of the study are stated at the end of the introduction but can seem misleading to the reader. The authors should put some effort into making the aims completely unambiguous to the readers. Most importantly, the different between the pilot and the main study needs to be very clear. 

We have restated the objectives as such, The objective of the current pilot randomized trial is in determining if the intended processes of the definitive study (i.e. the recruitment of participants with opioid use disorder, the process of randomization into two separate intervention arms, and the administration of the intervention) would work.”  

The very last paragraph of the introduction - which aims to justify why a pilot study is conducted - could be better integrated into the text instead of appearing as a separate paragraph at the end. 

Thanks for this suggestion. We have included this within the aim/objectives paragraph.

Methods

The lower age limit (21 years) should be explained.

We seek to clarify that we wish to recruit only adults into this study. In any case, the inpatient unit at the National Addictions Management Service only caters to adult patients.  

A power calculation has to be included, and needs to be of relevance both to the pilot study and to the main study. 

We are of the opinion that a power computation would be most essentially for the actual definitive randomized trial. We have proposed a sample size of 30 participants, with 15 participants each randomized to the conventional and gamified application respectively. As the evaluation of mobile attention bias modification intervention is new and novel, we have no other comparative studies to guide us in the computation of the sample size and the determination of the sample size necessary for this pilot study. In our scoping review of web-based cognitive bias interventions (http://www.ncbi.nlm.nih.gov.remotexs.ntu.edu.sg/pmc/articles/PMC6915808/), we have reported there being 2 pilot randomized controlled trials for obsessive-compulsive disorder. The maximum sample size was between 6 to 37 participants in each arm. We are also guided by our prior feasibility study, in which we managed to recruit 30 participants over the course of one year. In addition, based on Bell LM et al. (2018)’s (24) guidance for using pilot studies in informing the design of confirmatory studies, the proposed sample size for the pilot study if the effect size is small (i.e. Cohen D is between 0.1 to 0.3) is that of an estimated 25 participants per arm of the intervention. Hence, we are of the opinion that our proposed sample size of 30 participants in total is adequate, as we have followed the general rule of thumb. In addition, we have not performed a power computation for the purposes of the pilot, as the pilot study is supposed to inform the definitive sample size for the confirmatory trial. This was been advocated for in prior research: (A) WhiteheadAL , Julious SA , Cooper CL , et al Estimating the sample size for a pilot randomised trial to minimise the overall trial sample size for the external pilot and main trial for a continuous outcome variable. Stat Methods Med Res 2016;25:1057–73.doi:10.1177/0962280215588241 (b) BillinghamSA , Whitehead AL , Julious SA. An audit of sample sizes for pilot and feasibility trials being undertaken in the United Kingdom registered in the United Kingdom Clinical Research Network database. BMC Med Res Methodol 2013;13:104.doi:10.1186/1471-2288-13-104

In the randomization section, the authors refer to the work of the author 'Simon S'. Is that name correct?

We have changed it to SD Simon (1999). The reference remains unchanged.

It is difficult to understand whether participants will be informed and included at the first assessment or at the end of the detox period? Will the baseline measure take place prior to informed consent? The exact time schedule should be clarified. 

We seek to clarify that informed consent will be taken upon screening, on Day 7 of the detox period. We have clarified this in the experimental procedure and data collection. Figure 2 has also highlights that informed consent will be taken at baseline. The baseline measures will be administered on Day 8. The schedule of task administration have bene reflected in Figure 2.

The authors may need to justify why religion is assessed as one of the variables collected from patient, or consider removing this from the study. To me, it is of unclear relevance, unless the authors argue otherwise. 

We have decided to collect religion as one of the baseline demographic data. This variable might be of importance in further data analyses, and for the qualitative component of this manuscript, when describing the characteristics of the participants.

The authors should consider naming the 'perspective questionnaire' in a better way. Is this a questionnaire measuring self-reported or subjective patient experience, patient satisfaction or similar? Is this questionnaire based on any previous study, any previous data or an evidence-based measure? Please explain this. 

We have expanded and include the terminology self-reported questionnaire. This questionnaire has been previously utilized for the collation of patients’ perspective in our prior feasibility and acceptability study. (PMID:31586359)

The authors refer to a qualitative sub-study, but as I understand it, it is not included in the present study protocol? Please clarify this. 

We seek to clarify that the qualitative sub-study is based on the acquired participants’ perspectives captured using the perspective questionnaire. We will analyse these perspectives qualitatively.

Reviewer 3 Report

The manuscript describes the design of a non-blinded pilot randomized control trial to investigate the effectiveness of a conventional versus a new gamified/co-designed application to treat opioid use disorders. The paper is generally well-written although there are some minor grammatical errors. Some concerns are below:

1. The possibility of selection bias occurring during the study should be more thoroughly discussed.

2. A more thorough justification should be given for the sample size selected  for the study. It does not appear that a power analysis was performed to justify the sample size.

3. More discussion is needed on the generalizability of the expected outcomes.

Author Response

Replies to Peer Reviewer 3

We thank you Peer Reviewer 3 for your comments and recommendations. Please find in-line our responses to your comments.

The manuscript describes the design of a non-blinded pilot randomized control trial to investigate the effectiveness of a conventional versus a new gamified/co-designed application to treat opioid use disorders. The paper is generally well-written although there are some minor grammatical errors.

We have checked through the manuscript and have addressed this accordingly.

Some concerns are below:

The possibility of selection bias occurring during the study should be more thoroughly discussed.

- We thank you for highlighting this. We acknowledge that selection biases might be present in our study. We have included further information, as follows “In this current pilot study, we acknowledge that selection biases are present, in that we are selecting a cohort of participants that voluntarily seek admission to the inpatient unit. Moreover, the administration of the screening test would also imply that we are selecting specifically for individuals only with positive baseline biases. We are of the opinion that such biases could not be reduced and removed, and we will acknowledge them in our final published study. We will also acknowledge that such biases might affect the overall generalizability of our results.”

.

A more thorough justification should be given for the sample size selected  for the study. It does not appear that a power analysis was performed to justify the sample size.

- Reviewer 2 has also highlighted this. Please find as enclosed our responses:

We are of the opinion that a power computation would be most essentially for the actual definitive randomized trial. We have proposed a sample size of 30 participants, with 15 participants each randomized to the conventional and gamified application respectively. As the evaluation of mobile attention bias modification intervention is new and novel, we have no other comparative studies to guide us in the computation of the sample size and the determination of the sample size necessary for this pilot study. In our scoping review of web-based cognitive bias interventions (http://www.ncbi.nlm.nih.gov.remotexs.ntu.edu.sg/pmc/articles/PMC6915808/), we have reported there being 2 pilot randomized controlled trials for obsessive-compulsive disorder. The maximum sample size was between 6 to 37 participants in each arm. We are also guided by our prior feasibility study, in which we managed to recruit 30 participants over the course of one year. In addition, based on Bell LM et al. (2018)’s (24) guidance for using pilot studies in informing the design of confirmatory studies, the proposed sample size for the pilot study if the effect size is small (i.e. Cohen D is between 0.1 to 0.3) is that of an estimated 25 participants per arm of the intervention. Hence, we are of the opinion that our proposed sample size of 30 participants in total is adequate, as we have followed the general rule of thumb. In addition, we have not performed a power computation for the purposes of the pilot, as the pilot study is supposed to inform the definitive sample size for the confirmatory trial. This was been advocated for in prior research: (A) WhiteheadAL , Julious SA , Cooper CL , et al Estimating the sample size for a pilot randomised trial to minimise the overall trial sample size for the external pilot and main trial for a continuous outcome variable. Stat Methods Med Res 2016;25:1057–73.doi:10.1177/0962280215588241 (b) BillinghamSA , Whitehead AL , Julious SA. An audit of sample sizes for pilot and feasibility trials being undertaken in the United Kingdom registered in the United Kingdom Clinical Research Network database. BMC Med Res Methodol 2013;13:104.doi:10.1186/1471-2288-13-104

More discussion is needed on the generalizability of the expected outcomes.

- We acknowledge that there might be selection bias in our study. The presence of selection bias might affect the generalization of our outcomes measures.

Round 2

Reviewer 1 Report

The authors have made efforts to improve the manuscript and respond to reviewer's comments. The manuscript is now definitely improved, but I have a minor point.

The aims om p 4 first 4 lines of the yellow text ought to be presented at the end of the § to make it easier to follow.

Author Response

Dear Reviewer 1, 

We thank you for your acceptance of our revisions. 

We have inserted a new paragraph for the aims in our revised manuscript. 

Best,

Melvyn

Reviewer 3 Report

The authors have improved the manuscript after the revisions. I have no further recommendations. 

Author Response

Dear Reviewer 3, 

We thank you for your kind comments.